# Evaluating the Readiness of Ships and Ports to Bunker and Use Alternative Fuels: A Case Study from Brazil

Huang Wei [1], Eduardo Müller-Casseres [1], Carlos R. P. Belchior [2] and Alexandre Szklo [1,*]

1   Centre for Energy and Environmental Economics (CENERGIA), Centro de Tecnologia, Sala C-211, Cidade Universitária, Ilha do Fundão, Rio de Janeiro 21941-972, RJ, Brazil; huangkenwei@ppe.ufrj.br (H.W.); ecasseres@ppe.ufrj.br (E.M.-C.)
2   Ocean Engineering Program—COPPE/UFRJ, Centro de Tecnologia, Sala C-203, Cidade Universitária, Ilha do Fundão, Rio de Janeiro 21945-970, RJ, Brazil; belchior@oceanica.ufrj.br
*   Correspondence: szklo@ppe.ufrj.br; Tel.: +55-21-3938-8760

**Abstract:** The International Maritime Organization (IMO) has recently revised its strategy for shipping decarbonization, deepening the ambition to reduce annual greenhouse gas emissions by 2050. The accomplishment of this strategy requires the large-scale deployment of alternative maritime fuels, whose diversity and technical characteristics impose transition challenges. While several studies address the production of these fuels, a notable gap lies in the analysis of the required adaptations in vessels and ports for their usage. This study aims to fill this gap with a comprehensive review of material compatibility, storage in ports/vessels, and bunkering technology. First, we analyze key aspects of port/vessel adaptation: physical and chemical properties; energy conversion for propulsion; fuel feeding and storage; and bunkering procedures. Then, we perform a maturity assessment, placing each studied fuel on the technological readiness scale, revealing the most promising options regarding infrastructure adaptability. Finally, we develop a case study from Brazil, whose economy is grounded on maritime exports. The findings indicate that multi-product ports may have the potential to serve as multi-fuel hubs, while the remaining ports are inclined to specific fuels. In terms of vessel categories, we find that oil tankers, chemical ships, and gas carriers are most ready for conversion in the short term.

**Keywords:** alternative fuels; port; ship; bunker; biofuels; LNG; ammonia; methanol

## 1. Introduction

Maritime Transport is a key sector of the global economy, accounting for approximately 90% of the global trade in mass basis [1,2]. Shipping is a fundamental mode of trade for consuming less fuel per mass transported and distance covered compared with alternative modes. According to the Fourth IMO (International Maritime Organization) GHG (greenhouse gas) Study [3], the shipping world fleet consumed 13.6 exajoules (EJ) in 2018 and emitted 1.056 billion tonnes of carbon dioxide equivalent ($CO_2$eq), being responsible for nearly 3% of global greenhouse gas emissions. International shipping was responsible for 87% of the total emissions. Smith et al. [4] suggest that in the absence of measures to reduce greenhouse gas emissions, these emissions could increase by 250% by the year 2050. Among the available strategies to mitigate such emissions is to set speed, power, and fuel consumption limits [5]. Conversely, the vast diversity of ship types, with its associated challenges in construction and operation, has been a great barrier to standardization [6], in addition to the long lifetime of long-distance ships. Several studies have evidenced that the implementation of measures and technologies targeting a reduction in greenhouse gases (GHG) holds the potential to curtail emissions by up to 75% of the current levels [7–9].

In 2023, IMO established a goal of achieving net zero GHG emissions[1] by 2050, accounting for the life cycle emissions of fuels, while a medium-term goal entails achieving a minimum 20% reduction in GHG emissions from international shipping by 2030, as compared with emissions levels recorded in 2008 [11]. This new strategy exhibits a

greater degree of firmness when contrasted with IMO's initial and ambitious approach, which primarily focused on a reduction in shipping direct GHG emissions by a minimum of 50% in relation to 2008 levels [12]. Smith et al.'s [4] estimation indicated that the shipping sector emitted 921 million tons of carbon dioxide ($CO_2$) by 2008. According to DNV GL [13], to achieve previous IMO 2050 goals, it was imperative that 40% of the energy supplied to the shipping fleet was derived from fuels characterized by net zero emissions in ships. Faber et al. [3] predicted that without intervention, emissions could escalate to over 1300 million tons of $CO_2$ by 2030 and surpass 2300 million tons by 2050. Consequently, in comparison with a scenario with no actions to lessen the emissions, a decrease of more than 560 million tons of $CO_2$ emitted would be necessary by 2030.

To mitigate GHG emissions [14], several measures can be used, but the utilization of fuels with lower emissions levels or net zero emissions throughout their life cycle will be required [15]. The 2023 IMO guidelines on the removal of regulatory barriers concerning the blend of marine fuels with up to 30% of alternative fuels, specifically biofuels or synthetic fuels, encompass a fundamental factor in promoting the entrance of these alternative fuels into the shipping market. The blends with alternative fuels are to be treated on par with regular fuels, implying that they can be utilized as long as they comply with NOx emission limits [16,17].

Therefore, the investigation of alternative fuels for maritime transport has earned significant interest from both the academic and professional community. Recently, there has been a substantial number of studies delving into the subject of the production and consumption of biofuels [18–22], hydrogen and ammonia [23–27], liquefied natural gas (LNG) [28–30], and methanol [31–34] for shipping. While a significant share of these studies focuses on the technical aspects of production [35–41], emissions mitigation [7,42–44], and their consumption in marine engines [45–47], few have given due attention to the necessary adaptations required in ships and ports to the operation of these alternative fuels. Actually, the implementation of alternative fuels in the maritime sector drives various adjustments within ships. These modifications encompass alterations in fuel tanks and engine locations, utilization of distinct materials for storage tanks and pipelines, reinforcement of pipe structures, enhancement in ventilation systems to mitigate potential gas leakage [48], and changes in port infrastructure.

Therefore, the primary aim of this study is to assess the current progress of adjusting ships and ports to effectively use selected alternative fuels, with a particular emphasis on their applicability to long-haul cargo shipping, mostly characterized by large vessels, which significantly contributes to the sector's overall energy demand and GHG emissions [3]. By doing so, this analysis seeks to determine the technological readiness for the conversion of ports and ships to the storage, bunkering, and use of the chosen fuels. Some of the highlighted fuels can have their production based on both fossil and sustainable sources. For instance, LNG, methanol, ammonia, and hydrogen can be produced from fossil fuels and biomass or with electrolysis and carbon capture and storage (CCS) and direct air capture (DAC), known as e-fuels [49]. Both bio and e-fuel alternatives possess the potential to reduce GHG emissions when compared with fossil-based fuels [43]. Given that the primary goal of this analysis is to assess the compatibility of alternative fuel handling, storage, and usage, our discussion does not encompass an evaluation of the GHG emissions of these fuels from their production to their consumption. This has been completed in several works, such as Muller-Casseres et al. [35] and Brynolf et al. [39].

In addition, since this study addresses long-distance freight transportation based on large vessels, energy carriers and storage options, such as hydrogen and batteries, are not evaluated given their low suitability for deep-sea large ships, as shown by Gray et al. [50] and Xing et al. [40]. Indeed, these alternatives lead to a substantial spatial allocation loss in comparison with conventional fuels, making them impractical for long-distance shipping [51].

Then, to validate and illustrate the assessment conducted in this study, a case study was carried out to assess the capacity of the Brazilian fleet and port infrastructure to adopt alternative fuels. The Brazilian case is emblematic since the country's economy heavily

relies on marine routes [52] for exporting goods and sustaining its economic activities [53]. Additionally, Brazil has an impressive potential for alternative fuel production, particularly biofuels, given its abundant availability of biomass resources and established expertise in biofuels production [54]. For instance, according to Carvalho et al. [37], the comparative analysis encompassing Brazil, Europe, South Africa, and the USA illustrates that "biomass concentration in Brazil makes it the region with highest biobunker potential, which are mostly close to coastal areas and surpasses regional demand".

The next section outlines the methods and materials used in the evaluation. In Section 3, the results of the analysis are presented, focusing on determining and comparing the readiness of each alternative fuel. Section 4 delves into a comprehensive discussion of previous findings by applying them to a specific case study. Lastly, Section 5 provides the conclusions, along with recommendations and barriers identified in this study.

## 2. Materials and Methods

The primary objective of this study is to analyse the necessary adaptations in large deep-sea ships and ports for the proper storage, transfer, and utilization of alternative marine fuels. As such, it does not encompass fuels that can be classified as fully drop-in [49], such as Fischer–Tropsch liquids [38,55] from biomass and electric-derived hydrogen and $CO_2$. The deployment of these drop-in fuels can rely on existing ships and bunkering infrastructure, thereby enabling a direct replacement or blend with conventional fuels [56]. In contrast, most candidate alternative marine fuels require some level of adaptation in ships and ports. Some of them can be seen as partially drop-in, meaning that they only require minor adjustments and specific attention compared with conventional fuels to be used in the existing infrastructure. On the other hand, a second group (non-drop-in fuels) require substantial changes and investments in vessel technology and bunkering infrastructure. As this paper will discuss further, some of these non-drop-in fuels already have an established infrastructure in several ports (for example, the case of tradeable ammonia and methanol) and have a relevant usage record in dual engines (LNG and, to a lesser extent, methanol). However, the categorization here considers that more than 95% of large ships are still based on diesel engines and the ports associated with their routes are mostly single hubs to store and bunker petroleum-derived fuels for them [4]. This study focuses on the assessment of specific fuels encompassed by these two categories, as listed in Table 1. As mentioned before, it is worth noting that ammonia, LNG, and methanol can be produced from fossil, bio, and synthetic feedstocks. Our focus here is not on their production but on their handling and usage.

**Table 1.** Fuel grouping.

| Partially Drop-In [1] | Non-Drop-In [1] |
| :---: | :---: |
| Biodiesel | Ammonia |
| Hydrotreated pyrolysis oil (HPO) | Liquefied natural gas (LNG) |
| Hydrotreated vegetable oil (HVO) | Methanol |
| Straight vegetable oil (SVO) | |

[1] [19,35,57].

A comprehensive and thorough review of the technical literature was conducted, with a specific emphasis on the essential properties to be taken into consideration for achieving a successful adaptation in retrofitting both ships and ports to enable proper storage, transfer, and utilization of alternative fuels. Figure 1 provides a summary of the steps undertaken in this study. This analytical study first examined various aspects pertaining to selected alternative fuels. As a second step, considering the existing ships and bunkering infrastructure globally, along with regulatory frameworks and tests designed to assess fuel performance on ships, the analysed fuels were categorized into those that are partially or non-drop-in. This categorization was succeeded by an assessment of technological readiness based on the guidelines provided by the US Department of Energy [58].

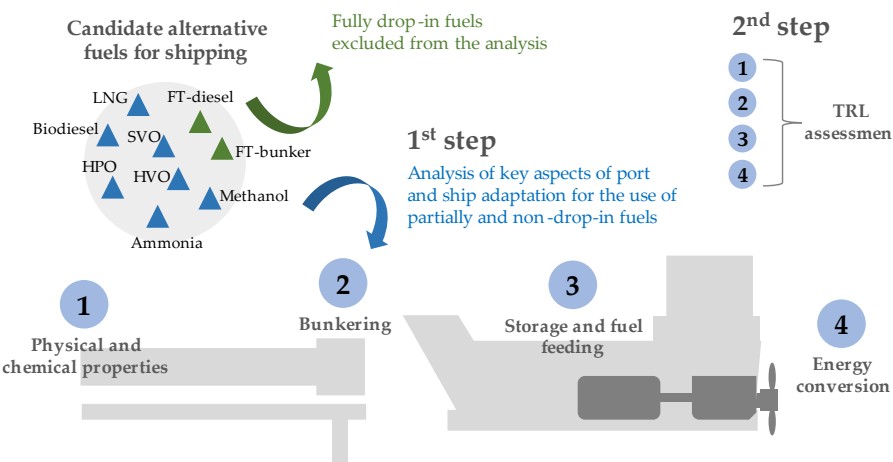

**Figure 1.** Methodological procedure.

As Figure 1 displays, the first step in the analysis encompasses the key aspects of port and ship conversion for the proper utilization of the selected alternative fuels. The first step was split into four main aspects, namely, physical, and chemical characteristic properties, bunkering procedures, storage and fuel feeding systems, and energy conversion systems. Table 2 displays the main aspects analysed for each of the aforementioned segments.

**Table 2.** Main aspects analysed for each section concerning the adaptation of ships and ports to the use of partial and non-drop-in fuels.

| Segment | Analysed Aspects |
|---|---|
| Physical and chemical properties | Heating value<br>Volumetric density<br>Energy density<br>Kinematic viscosity<br>Acidity<br>Flash point<br>Self ignition temperature<br>CCAI<br>Other properties |
| Bunkering | Pressurization<br>Liquefaction<br>Tank shape<br>Inertisation<br>Ventilation<br>Maintenance |
| Storage and fuel feeding | Pressurization<br>Liquefaction<br>Tank location<br>Tank volume<br>Inertisation<br>Ventilation reinforcement<br>Maintenance<br>Need for double-wall<br>Materials<br>Drainage<br>Preheating<br>Filtering |
| Energy conversion | Converter type<br>Need for pilot fuel<br>Engine adjustments |

As Table 2 illustrates, the initial analysis includes a review of the main properties of fuels in comparison with conventional fossil bunker fuels. Heating value and volumetric density are both linked to energetic density, which represents the amount of energy per cubic meter. In shipping, greater energetic density is preferable as it allows for increased autonomy due to the higher energy demand for fuels (e.g., Ref. [59]), as well as smaller losses of freight space [50]. High levels of kinematic viscosity directly impact the spray and flow characteristics of fuel [60]. Acidity is associated with the content of free fatty acids in fuel. A high content of free fatty acids can result in engine deterioration, as well as degradation of engine feed [61]. Flash point refers to the minimum temperature at which gases ignite when exposed to a flame [62]. Hence, low-flash point fuels are undesirable for shipping. Ellis and Tanneberger [31] underscored that low flash points trigger additional safety measures in order to prevent the fuel from being exposed to ignition sources. A high self-ignition temperature leads to obstacles in achieving auto-ignition, a characteristic considered unfavourable for use in diesel engines [63]. The aromaticity index, measured with the calculated carbon aromaticity index (CCAI), is used to assess fuel quality based on ignition delay. CCAI is calculated with an evaluation of density and viscosity. For marine engines, a CCAI below 870 is recommended [64]. Viscosity and CCAI values for LNG and ammonia are not evaluated in the literature since they are equivalent to or lower than those of traditional fuels. As a result, these factors were not considered in this study, nor were the acidity levels in LNG, methanol, ammonia, and HVO. Other properties, such as oxygen and water content, play a pivotal role in determining the requisite adjustments for utilizing these fuels in the current infrastructure.

Having addressed the main properties of fuels, this study evaluated the necessary adjustments to bunkering infrastructure to accommodate the usage of each selected fuel. As indicated in Table 2, certain aspects were examined, including the requirements for pressurization, liquefaction, different tank shapes, inertisation, ventilation reinforcement, and an increase in maintenance. This evaluation encompassed not only the bunkering process but also storage at ports.

Then, this study revised the challenges related to storage and fuel feeding in ships. The analysis carried out addressed significant modifications resulting from distinct properties of the chosen fuels, as opposed to conventional fossil bunker fuels. Aspects such as demands for pressurization and liquefaction during storage, different shapes, locations, and volumes of tanks, double walls, and filtering were highlighted.

Finally, the energy conversion analysis addressed the available choices of energy converters for each fuel, with a specific emphasis on a potential pilot fuel demand and adjustments in engines for the proper use of the fuels. The analysed options for energy converters are diesel engines, dual-fuel engines, and fuel cells. According to the Fourth GHG IMO Study [3], conventional fossil bunker fuels, namely, heavy fuel oil (HFO) and marine diesel oil (MDO), are the two primary fuels commonly used in the marine industry, representing 66.0% and 30.5% of the world's consumption, respectively. Additionally, LNG accounted for roughly 3.4% of the world's consumption, whereas methanol represented a mere 0.05% of the overall shipping consumption. As a result, the predominant energy converter to propulsion in the vessel fleet is the two-stroke diesel engine. In 2018, low, medium, and high diesel engines accounted for over 98% of the global marine fleet, while dual-fuel LNG engines were installed in less than 0.5% of ships, and engines adapted to methanol were reported in less than 0.15% of the fleet [3]. Diesel engines designed for marine applications are available in two configurations: two- and four-stroke variants. Larger ships typically opt for two-stroke engines due to their ability to achieve lower propulsion speeds effectively. In contrast, medium- and high-speed engines predominantly use four-stroke cycles to optimize the operation of these vessels [65].

In relation to the conversion of diesel engines to dual-fuel engines, Tiwari [66] reported that the dual-fuel engine is essentially a diesel engine equipped with supplementary devices that enable the utilization of fuels such as LNG. Bhavani and Murugesan [67] further pointed out that the conversion from diesel to a dual-fuel mode solely necessitates external modifications to the engine, while the internal components remain unchanged. Furthermore, the authors emphasized that the conversion process involves the addition of a set of retrofit components, including fuel supply systems, pilot and supplemental fuel inlet controllers, air and gas mixers, engine cooling systems, flameproof kits, and gas detectors. Another viable energy converter option is the use of fuel cells, which are currently in the developmental phase for marine applications. Nevertheless, fuel cells present superior efficiency and emit fewer pollutants during tank-to-wake, namely, the use in ships, when compared with internal ignition and gas engines. In addition, a steam reformer can be incorporated into vessels to enable the use of hydrocarbons as an energy vector. Although this process generates carbon dioxide emissions, they are significantly lower than those produced by conventional engines utilizing fossil fuels, and the emissions of other pollutants remain nearly negligible [68]. Xing et al. [51] stated that recent research and demonstration projects have validated the technical feasibility of fuel cells for maritime applications regarding power capacity, safety, durability, and operational terms. These developments contribute to promoting the adoption of fuel cells in vessel fleets in the future. However, it is important to note that despite these advancements, commercial viability remains a challenge [46], and their suitability for long-haul shipping is still limited [51].

Having addressed all segments of the first step, the evaluation of TRL for each fuel is thus complete. Figure 2 summarizes the assessment approach.

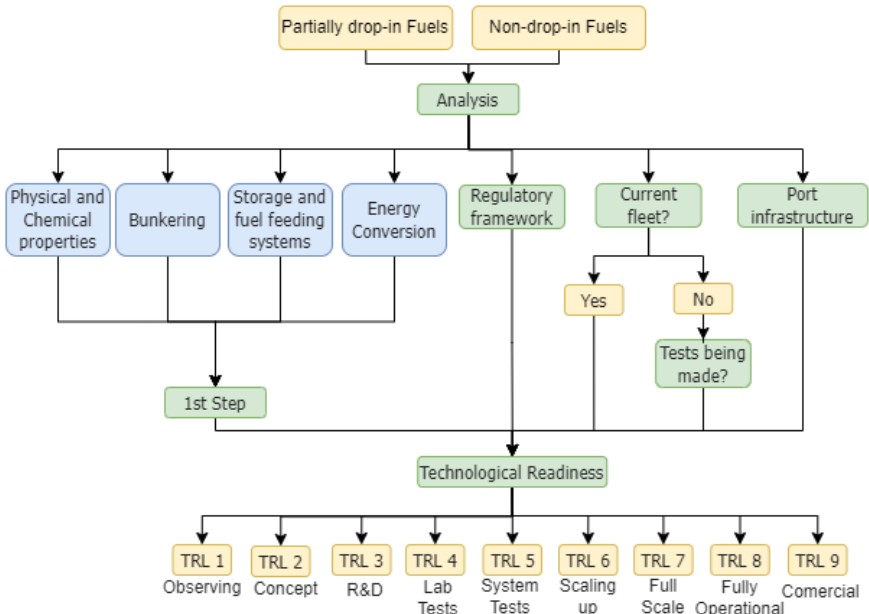

**Figure 2.** TRL evaluation of each fuel type.

As Figure 2 illustrates, the determination of TRL for each fuel resulted from the analysis performed, also considering the current regulatory and port infrastructure. A detailed exploratory review was performed to identify the established standards, guidelines, and whitepapers conducting procedural aspects associated with the utilization of each designated fuel, thus enabling an assessment of the regulatory framework. Current infrastructure evaluation was also performed by compiling data pertaining to vessels that already adopted alternative bunker fuels. In the absence of ships using fuel, a review encompassing not only vessels but also other modes of transportation were conducted. Furthermore, an evaluation of port infrastructure was conducted to identify existing port facilities offering bunkering services for each fuel. The required adjustment to each fuel for

use in maritime infrastructure facilities leads to the estimation of TRL. This ranges from observation of technology (TRL 1) to conceptualization (TRL 2), research and development or R&D (TRL 3), laboratory tests (TRL 4), systems tests in real conditions (TRL 5), scaling up in real conditions tests (TRL 6), full scale in real conditions tests (TRL 7), and fully operational functioning (TRL 8) to reach commercial status (TRL 9) [58].

Finally, after conducting a comprehensive assessment of the obstacles and complexities involved in adapting the existing maritime infrastructure to accommodate alternative fuels, this study applied it to a case study as a representative example. The case study was based in Brazil, given its high economic dependency on maritime routes, from cabotage to national trade and long-haul distances for exportation [52,53], as well as its notable potential as a major future biobunker producer [37]. The case study examined the current state of the Brazilian shipping sector, including high-priority ports, given their cargo movement and initiatives to bunkering of alternative fuels; an analysis of potential multi-fuel hubs; the progress and challenges in converting ships for alternative fuels; the initiatives assumed by local governments and companies linked to the maritime sector to achieve decarbonization of Brazil's maritime transport; thermal stability of fuels in maritime routes; and the problem of loss in cargo space. The primary objective was to develop a coherent framework that would evaluate the potential of introducing alternative fuels in the country. This framework can serve as the first roadmap for assessing the feasibility of applying alternative fuel solutions in Brazil and potentially extrapolate these findings to other countries and regions with similar characteristics.

## 3. Results

### 3.1. Physical and Chemical Properties

Table 3 lists the main properties of the selected alternative fuels.

**Table 3.** Properties of marine fuels.

| Fuel Property | Heating Value | Volumetric Density | Energy Density | Viscosity at 40 °C | Acidity | Flash Point | Self-Ignition Temperature | Aromaticity Index (CCAI) |
|---|---|---|---|---|---|---|---|---|
| Unit | MJ/kg | kg/m$^3$ | MJ/m$^3$ | mm/s$^2$ | Mg KOH/g | °C | °C | - |
| HFO | 40.0 [a] | 991 [a] | 39,640 | 380 [i] | 2.5 [i] | >60 [i] | 407 [P] | 856.5 [u] |
| MGO | 42.0 [a] | 890 [a] | 37,380 | 3.5 [i] | 0.5 [i] | >60 [i] | 257 [q] | 808.1 [u] |
| LNG | 50.0 [b] | 415 [b] | 20,750 | - | - | −188 [b] | 537 [o] | - |
| Biodiesel | 37.1[c] | 885 [c] | 32,833.5 | 4–6 [j] | 0.052–0.295 [m] | >93 [c] | 374–449 [r] | 822.6 [u] |
| SVO | 37–39.62 [a] | 900–930 [a] | 33,300–36,847 | 14–40 [k] | 0.02–20 [n] | >400 [k] | 405 [s] | 836.6–878.7 [u] |
| HVO | 44.1 [d] | 780 [d] | 34,398 | 3 [d] | - | 99 [d] | 204 [o] | 738.4 [u] |
| HPO | 28.9 [e] | 1150 [h] | 33,235 | 9 [h] | 21.3–76.1 [h] | 53–101 [h] | 340 [t] | 1076 [u] |
| Ammonia | 18.6 [g] | 758 [g] | 14,101 | - | - | 132 [o] | 630 [o] | - |
| Methanol | 20.1 [f] | 798 [f] | 16,040 | 0.58 [l] | - | 12 [f] | 470 [o] | 837.6 [u] |

a—[62]; b—[69]; c—[18]; d—[70]; e—[71]; f—[72]; g—[73]; h—[74]; i—[75]; j—[76]; k—[77]; l—[31]; m—[78]; n—[79]; o—[57]; p—[80]; q—[81]; r—[82]; s—[83]; t—[84];u—[85].

In the comparison among fossil fuels, LNG stands as the option for the mitigation of sulphur oxides, nitrogenous oxides, and particulate matter emissions [86]. It is predominantly composed of methane, accompanied by minor proportions of other hydrocarbons such as ethane, propane, and butane [29]. Under atmospheric temperature and pressure, LNG is in the gaseous phase and has low density. In order to optimize storage, natural gas is liquefied at a temperature of −162 °C and atmospheric pressure, thereby reducing the required volume for storage [69].

The properties of biofuels vary depending on the feedstock used in production. Biodiesel, SVO, HVO, and HPO have energy density levels close to HFO and MGO compared with the other assessed fuels, suggesting that those fuels have greater potential to provide increased autonomy or reduced storage space requirements. SVO is a biofuel that entails a straightforward production process in comparison with other fuels. The production steps involve biomass collection, low-temperature seed pressing, and filtration to remove sludge. The quality of the fuel is heavily influenced by the quality of the feedstock and the conditions during production and processing [87]. When contrasted with traditional marine fuels, SVO has a slightly lower energy density and a higher flash point, viscosity, and acidity. These characteristics can potentially result in corrosion of engine feed pipelines [62]. Biodiesel (or FAME), widely regarded as one of the most promising biofuels, is repeatedly stated as a potential blend component for diesel in the road transport sector [88].

HVO consists of straight chains of paraffinic hydrocarbons, which undergo additional production steps in comparison with SVO. These steps include catalytic saturation (hydrogenation), hydrodeoxygenation, hydrodecarboxylation, and isomerization. HVO is distinguished by its exceedingly low sulphur content and minimal emission factors [70]. As a paraffinic compound, HVO exhibits a high cetane number, typically ranging from 75 to 95 [89].

Hydrotreated pyrolysis oil, also known as bio-oil or HPO [90], is derived from biomass, which undergoes a high-temperature process in the absence of oxygen. The biomass is subjected to a temperature of 500 °C for a brief duration [21]. Hydrogenation is the final step, transforming the pyrolysis oil into hydrotreated pyrolysis oil. Depending on the pyrolysis process, the water content in bio-oil can reach up to 30%, which is sufficient to induce phase separation when stored at ambient temperature for six months [20]. Treatment of bio-oil can result in a compound with a significant reduction in oxygen content and an increase in light aromatic compounds.

In relation to viscosity, SVO and HPO have elevated levels, necessitating appropriate measures to viscosity decrease such as preheating. Moreover, these fuels are also notable for their high acidity levels. Biodiesel has a viscosity greater than traditional diesel yet not as high as SVO and HPO; therefore, preheating is advisable [22]. HPO has a high and unstable viscosity, posing a challenge for both its use as a fuel and storage [91]. Notably, the low flash point of biodiesel limits its practical utilization in low air temperature conditions [45]. HVO has a flash point higher than traditional fuels [89].

The acidity level of SVO, as is the case for biodiesel, is associated with its specific feedstock, which is also the case for biodiesel. While certain vegetable oils may present higher acidity levels compared with HFO, others exhibit relatively low acid values, as exemplified by rapeseed oil, which has an acidity level below 2.5 mg KOH/g [87]. Despite undergoing a reduction of approximately 70% in acidity after treatment, the resultant HPO acidity level remains notably higher when compared with traditional marine fuels [74].

The majority of the discussed fuels exhibit an aromaticity index below the recommended limit. However, depending on the feedstock used, the aromaticity index of SVO may exceed the suggested limit, as is the case for HPO. Ellis and Tanneberger [31] draw attention to the possibility of utilizing a lubricant oil to address the issue of low lubricity. In comparison with traditional fuels, biodiesel has superior lubricity and lower toxicity levels. However, it possesses a high oxygen content, typically ranging between 10 and 11%, and a low pour point [45,47,62]. To mitigate the risk of corrosion, the usage of a corrosion inhibitor known as tert-butylamine is advisable, with a recommended concentration of 250 ppm [47].

Methanol [92] and ammonia [93] are widely used as feedstocks in the chemical industry. Given their high toxicity, it is essential to implement safety measures to prevent leaks and human exposure to these substances, such as gas detectors. As stated by Kay et al. [94], ammonia leakage not only into the air, but also into the sea, can lead to critical damage, and lethality can be greatly reduced if the release duration is shortened. Overall, the authors

found that a 30 s leakage is 70% less lethal than a 60 s leakage. Safety measures must be targeted to mitigate toxicity, especially at potential sources of leakage such as inlet and outlet manifolds for hose connection. According to Hansson et al. [27], the presence of high concentrations of ammonia poses health risks and can prove lethal within certain concentrations and exposure durations.

Ammonia has been proposed as a potential sustainable energy carrier of hydrogen due to its composition of three hydrogen atoms per ammonia molecule ($NH_3$) [95]. In addition, the storage of liquid hydrogen requires extremely low temperatures, specifically, $-253$ °C [96]. Hydrogen is recognized as a promising marine fuel, with ongoing tests aimed at advancing its utilization in the shipping industry. However, as reported by ABS [97], hydrogen currently offers a very limited power output, associated with substantial costs and limited production. Additionally, hydrogen storage in vessels addresses significant problems that marine communities have yet to overcome. Kim et al. [73] also highlight that ammonia possesses 1.7 times higher energy content compared with hydrogen, along with a 50% greater hydrogen content by volume [26], leading to a reduced volume requirement of fuel storage. Alongside LNG, ammonia also necessitates lower temperatures and pressurization to maintain its liquid state during storage. Ammonia can be stored at 25 °C when pressurized at 10 bar, whereas under atmospheric pressure, the required storage temperature is $-33.4$ °C [73]. For ammonia, refrigerated storage is preferable due to its better effectiveness in reducing operational risk [94]. Methanol and LNG are low-flash point fuels, making them highly flammable. Methanol is flammable and exhibits lower lubricity compared with conventional marine fuels [31]. The flammability of ammonia [98], methanol [31], and LNG necessitates safety protocols to prevent the risk of leaks and spills, particularly in areas where ignition sources are present [99]. Regardless of its high flash point, ammonia has lower flame velocity compared with conventional fuels [93].

*3.2. Bunkering*

The bunkering of conventional fuels can be carried out using tank trucks (truck-to-ship-transfer or TTS), bunker vessels (ship-to-ship or STS), as well as shore tanks or pipelines (shore tank-to-ship or TPS) [100]. Regarding alternative fuels, the three aforementioned methods can be applied for bunkering, with specific protocols designed for each fuel type based on its distinct characteristics.

When using LNG bunkering, a security protocol must be followed to avoid leakages of the fuel under cryogenic conditions. If materials such as steel come into contact with LNG, they tend to become fragile and may experience cracking. The procedures for leak prevention are as follows: checking the connection of the supply pipeline, inertization of the pipeline with nitrogen gas, cleaning the interior of the pipeline with vapour from liquefied natural gas at cryogenic temperatures, bunkering, cleaning the remaining LNG inside the pipeline with vapour from natural gas at cryogenic temperatures, inertization of the pipeline with nitrogen, and disconnection of the supply pipeline [101]. Aneziris et al. [99] asserted that the utilization of low-temperature pipelines, loading arms, and hoses is mandatory for LNG bunkering. Furthermore, the authors also highlighted that it is essential to acknowledge that extremely low LNG temperatures may pose a significant hazard, impacting not only the structural integrity of materials used by causing potential cracks but also the safety of individuals in proximity to the LNG due to the risk of frostbite.

To ensure the appropriate bunkering of biofuels, it is imperative to modify storage tanks in accordance with specific fuel properties [19]. Ideally, the tanks should possess a narrow shape, aiming to reduce the retention of oil and fats during the cleaning process. Furthermore, the tank bottoms should be tapered to facilitate effective drainage [102]. The fuelling processes for SVO [62] and HVO [103] are comparable to those already established for HFO and marine diesel, respectively. However, as Kesieme et al. [62] stated, certain adjustments are necessary to safeguard against corrosion and water contamination. Additionally, the authors recommended that maintenance procedures should be reinforced to ensure prolonged use. Regarding HPO, the complete supply chain must be developed,

including the development of suitable bunkering infrastructure to accommodate the unique fuelling requirements of bio-oil [104].

All fuelling methods applicable to LNG can be used for ammonia as well. However, additional requirements must be met, specifically, the filling station must be equipped with appropriate ventilation, either using natural means or machinery. Additionally, the piping system must be self-draining and composed of inert materials [105]. Those adjustments are demanded due to the toxicity and flammability issues of ammonia, as previously addressed in Section 3.1 and extensively examined by Fan and Enshaei [98] and Kay et al. [94]. The latter study also recommended the use of multiple hoses with lower flow rates instead of a single hose with a higher flow rate, thereby resulting in a reduction in bunkering time and increased safety conditions. In order to ensure the appropriate bunkering of ammonia and prevent the release of the substance, it is imperative, as emphasized by Duong et al. [106], to develop a comprehensive strategy aimed at minimizing ammonia leakage during the bunkering process.

*3.3. Storage and Fuel Feeding*

Due to the low temperatures observed during storage, specific tanks become necessary when utilizing LNG in ships. Several options for storage tanks are available: IMO type A, which resembles the ones commonly used for standard marine fuel [107], IMO type C, designed as pressure vessels, and membrane tanks. Additionally, there exists a category called type B, encompassing all tanks that are neither type A, type C, nor membrane tanks. Among the mentioned tank types, type A and type B are the most suitable for larger vessels due to their generally prismatic shape [108]. However, an obstacle to the effective utilization of LNG as fuel is the occurrence of methane slip [86], which involves gas leakage during both storage and engine operation. This issue can be mitigated if the leaked gas is reclaimed and reused by other ship machinery, such as in gas combustion units [107]. Regarding engine fuel supply, in order to enhance LNG safety procedures, ABS [109] recommends the utilization of gas detection systems for instant shutdown, double-wall piping with at least 30 air changes per hour, a maximum 10 bar pressure limit, nitrogen-based inertization for emergencies, and independent pumps and compressors from other circuits.

The utilization of biofuels, such as biodiesel, necessitates the use of appropriate materials for tanks and pipelines. It is recommended that stainless steel, as a material, be used for this purpose. However, when the blends comprise no more than 20% biodiesel in the overall volume, conventional materials can be used if adequately coated with zinc. The construction of feed pipelines using mild steel is permissible, provided that filters are installed to ensure the smooth operation of the system [110]. Moreover, to maintain the integrity of the biofuel infrastructure and prevent any potential water contamination, regular and careful inspections, maintenance activities, and constant cleaning of tanks and piping are essential [102].

The coexistence of water within a fuel blend poses a significant risk of degrading fuel filter cartridges, potentially leading to cavitation [62]. To mitigate such hazards, the use of stainless steel is recommended as the material of choice for constructing pipelines and tanks to ensure optimal safety. Alternatively, mild steel can be considered for tank and pipeline construction if suitably coated with an inert material. However, it is imperative to conduct regular inspections of the tanks to assess the condition of the coatings and ensure their integrity is preserved. Furthermore, it is of utmost importance that all materials utilized in tanks and auxiliary machinery, including heating units, must be inert to vegetable oils [102].

HVO exhibits a great level of resemblance to conventional diesel-based fuels, rendering it compatible with the materials already used in marine infrastructure for pipelines, tanks, feed systems, and engines. Nevertheless, it is recommended to take on maintenance and cleaning procedures for storage tanks before fuelling to ensure optimal performance. Additionally, strict supervision is advised to prevent any contact between HVO and water within the tanks and feed system as this could lead to detrimental effects. Remarkably, HVO sets itself apart from other biofuels by causing minimal corrosion of the materials commonly

utilized in the naval industry's infrastructure. This exceptional property contributes to enhanced durability and safety in marine operations involving HVO usage [103].

The high viscosity characteristic of HPO leads to an increase in engine deposits, which subsequently requires more energy for pumping and results in accelerated wear on fuel pump components and injectors. To mitigate these effects, preheating the fuel is essential as it effectively reduces the viscosity level. For the engine feed system, it is imperative to construct it using corrosion-resistant materials to withstand the high acidity of the oil. Copper can be considered as a viable material option for tank storage and pipelines; however, it is recommended to utilize stainless steel for tanks and pipes. The high acidity of HPO poses limitations on the use of carbon steel in pumps, fuel lines, and burners. These components must be made of materials that can resist the corrosive nature of the fuel. Furthermore, due to the presence of solid particles with high energy density, filtering them is not considered desirable. Nevertheless, the careful design of fuel supply piping is essential to prevent any blockages resulting from solid particle materials. Moreover, both pumping and atomizing processes should be equipped with suitable filtration mechanisms to ensure smooth and efficient operation [104].

When utilizing fuels with high acidity and/or flammability in ships, it is imperative that fuel storage tanks in ships adhere to a double-walled construction for enhanced safety measures. These tanks can be positioned either at the main deck, offering a more economical and less complex installation, or at lower decks as long as they are sufficiently distanced and detached from accommodation and machinery spaces. To minimize the risk of gas leakages, stringent preventive measures must be used. These include the implementation of inert systems, reinforcement of ventilation, and the utilization of specialized materials such as aluminium or, preferably, stainless steel for storage, feed, and engine components [29]. Furthermore, it is crucial to ensure that the pressure within the feed system does not exceed 10 bar [109] to maintain operational safety. By sticking to these guidelines, the potential hazards associated with fuel storage and usage in ships can be effectively mitigated.

In 2020, IMO [105] issued a comprehensive set of guidelines regarding the utilization of methanol and ethanol in vessels, encompassing fuelling procedures and safety practices. Some of these practices were already disseminated by DNV GL. The recommended safety measures include the implementation of double-walled feed pipelines and storage tanks constructed from stainless steel or austenitic steel, the incorporation of inert gas purging devices to facilitate the controlled release of gas, the installation of service tanks with the capacity to power operational loads for a minimum of eight hours, and the use of high-pressure pumps with a minimum pressure of 10 bar to facilitate the fuel feed to engines [111]. It is preferable to position the service tank on the main deck, while the pilot fuel tank may be situated in the engine room [34]. Due to the highly toxic nature of methanol, all areas containing pipelines or tanks are required to have adequate ventilation reinforcement. Specifically, normal spaces require a minimum of 15 air renovations per hour, while spaces more susceptible to fuel leakage require 30 air renovations per hour [31].

Regarding the utilization of ammonia in vessels, the required tanks for storing ammonia should be pressurized, with a minimum pressure of 8.6 bar, while the recommended pressure level stands at 17 bar [112]. For optimal cost-effectiveness, the type C tank has demonstrated its superiority and versatility, as it can be conveniently installed on the main deck and seamlessly integrated into the majority of existing ships [113]. To ensure the safe handling of ammonia, the feed pipelines must be constructed using durable materials such as carbon and stainless steel [25]. These pipelines should be displayed in a double-walled configuration to mitigate the risks of leakage [111]. Additionally, it is mandatory to equip all spaces associated with the fuel storage system with a comprehensive ventilation system. This measure is indispensable in preventing any potential ammonia leakages [112], thereby enhancing overall safety and minimizing associated hazards. Additionally, with respect to fuel feeding, it is recommended to avoid corrosive materials such as copper, high-nickel alloys, and plastic. To prevent corrosion, it is advisable to use Teflon in engine seals instead of rubber and plastic [25]. A system for emergency ventilation must be installed and

operated in accordance with either of the following principles: a reduction in ammonia concentration to below 10 ppm with dilution or the capture of excessive ammonia [112]. In order to reduce the potential risks associated with ammonia leakage in the engine room, it is advisable to install both a tank and feed system on the deck, coupled with its connection to the engine using dual-walled piping. Another alternative is to place the feed system and tanks within the engine room if an airlock system to prevent ammonia dispersion on-site is installed [113]. DNV GL [111] suggested a mandatory provision of secondary enclosures for all fuel piping to securely contain any potential leaks. Furthermore, an arrangement involving the infusion of nitrogen into the secondary enclosure, coupled with ongoing pressure monitoring, can also be an alternative solution to ensure safety.

### 3.4. Energy Converters

The analysed fuels are applicable for one or more of the three energy converters considered in this study. With appropriate adjustments to adapt feed and combustion requirements, all fuels can be effectively applied in existing marine engines. Biodiesel [45], SVO [62], HVO [70], and HPO [91] demand relatively minor modifications to existing marine diesel engines and feed infrastructure. On the other hand, methanol and LNG, due to their high ignition temperature and consequently low cetane number, face ignition complications. To tackle this issue, dual-fuel engines can be used, in which a pilot fuel, such as marine diesel, is injected to start ignition [29].

Regarding SVO, to achieve the desirable viscosity levels, fuel preheating is imperative. The recommended heating temperature is within the range of 67 to 78 °C, which is comparatively lower than the temperatures required to preheat HFO [61]. Similarly, for the proper use of HPO in diesel engines, preheating within the temperature range of 40 to 80 °C is required [104]. It is crucial to be cautious of potential impurities in vegetable oils, since their presence may lead to engine failure or damage when used as marine fuel [62]. The combustion properties of HVO are similar to those of conventional fuels, such as marine diesel, although it has a lower density. Therefore, it is advisable to make adjustments in order to enable longer fuel injections for engine optimization, thereby increasing efficiency and fuel savings [70].

According to Dincer and Siddiqui [114], the use of ammonia in diesel engines presents drawbacks, notably, its limited flammability range, low kinetic rate, and high self-ignition temperature. Ammonia's combustion properties demand modifications to conventional combustion engines, as well as blending with fuels exhibiting superior combustion properties [115] and using dual-fuel engines. Burning ammonia may have the potential to produce more NOx emissions than regular fuels [116], potentially also releasing $N_2O$, a much stronger greenhouse gas than $CO_2$ [117]. Another alternative for ammonia is the use of fuel cells [114]. Kim et al. [73] compared the use of a polymer electrolyte membrane fuel cell (PEMFC), a low-cost alternative, and a solid-oxide fuel cell (SOFC) for ammonia chemical energy conversion. The results indicated that the latter is a simpler and more optimized operation for an ammonia-fuelled 2500 TEU container ship. SOFC used 12% less fuel on a volumetric basis.

### 3.5. Technological Readiness

The analysis by El-Gohary [118] demonstrated that the utilization of LNG as the primary fuel instead of conventional marine fuels has the potential to reach a notable reduction in annual expenses associated with fuel and maintenance, ranging from 30% to 40%. The implementation of LNG as the primary fuel for ships is rapidly becoming a reality. As of July 2023, a substantial portion of the global fleet, specifically 403 ships, has already adopted the use of LNG as fuel, and 275 terminals worldwide have equipped bunkering facilities for these vessels [119]. Consequently, the infrastructure for LNG bunkering has been firmly established, and all requisite fuel procedures have been meticulously documented by classification societies [109]. This thorough development and documentation

have led to the classification of LNG's practical use as commercially available, indicated by a TRL of 9.

Among all the analysed fuels in this study, only biodiesel was mentioned in standards until 2022, allowing its use in marine fuel blends. Specifically, ISO 8217:2017 enables the utilization of up to 7% $v/v$ of biofuel in such blends [120]. Mohd et al. [45] demonstrated that the direct use of biofuel in ships could potentially compromise current power supply systems, decrease efficiency and, consequently, increase specific consumption. However, Mohd et al. also pointed out that certain engine manufacturers, such as MAN, Wärtsilä and Caterpillar, have conducted tests showing satisfactory performance without necessitating modifications if the blend contains up to 30% $v/v$ of biofuel. Additionally, Ogunkunle and Ahmed [121] reported that blends containing 30% biofuel (B30) and diesel do not result in engine alterations, although there is an increase in specific consumption. Countless marine engine manufacturers have undertaken research and testing to enhance the implementation of biofuels in vessels. Despite this progress, the biofuel bunkering process in ships still requires further development, even though minor adjustments may be necessary [18]. Consequently, as a marine fuel, biofuel is still in the full-scale testing phase, awaiting validation under real operating conditions, characterized as TRL 7.

Kesieme et al. [62] asserted that although SVO and HFO share some similarities, it is improbable that a blend of these two types of fuels would be compatible. Consequently, the most practical and viable solution would be a complete replacement of HFO with SVO. The usage of SVO in marine applications is still under research, both as a drop-in replacement and as a blend with traditional fuels. It has been observed that if a blend contains no more than 20% $v/v$ of SVO with diesel, no changes in the fuel feeding systems of engines are necessary [122]. Furthermore, No [123] reported that a blend containing 20% $v/v$ of SVO and diesel does not require any alterations to the marine engine systems. Additionally, it was found that pre-heating SVO at temperatures ranging from 55 to 85 °C allows for an increase in the percentage of SVO in the blend by 30% to 60% $v/v$ without requiring changes in engine structures. Blin et al. [79] proposed that for drop-in usage of SVO in ships, a dual injection system should be used, where diesel would be injected at the start of the engine, and once it warmed up, SVO would be injected. The implementation of SVO as a marine fuel demands the development of a bunkering infrastructure [62], as well as further testing and refinement, leading to an assumed TRL regarding the use of the biofuel of 5.

HVO exhibits the potential to serve as a viable substitute for marine diesel, owing to its similar characteristics and compatibility with conventional ignition engines [123]. Currently, HVO is undergoing tests in the transport sector. Notably, numerous experiments have been conducted involving trucks and cars utilizing HVO either as a drop-in fuel or as a component in the fuel blend. These tests have been carried out in diverse countries, including Germany, Canada, the United States, Finland, and Sweden. One particularly significant test took place in the city of Alberta, Canada, demonstrating HVO's capability to function efficiently even in extremely cold temperatures reaching as low as −44 °C. However, despite investigations in road transportation, there was no documented record of HVO being tested in ships until the year 2022 [103]. Therefore, HVO emerges as the alternative marine fuel in this study, imposing the least modifications for its implementation in existing fleet and bunkering infrastructure. However, there exist certain barriers to the widespread adoption of HVO in the maritime sector, such as limited production capacity and high pricing, along with competition from the road and air sector [21]. To overcome these challenges and establish HVO as a viable marine fuel, further comprehensive studies and research are vital to assuring an assumed TRL of 5.

Concerning its utilization in marine engines, Chong and Bridgewater [90] stated that the blend of HPO with diesel and alcohol should not exceed 40% $v/v$. There is an emerging prospect that HPO may serve as a replacement for heavy oil in the future. However, its widespread adoption requires further research and comprehensive testing [104]. As a result of its early stage of development, HPO has been classified as having a low maturity level, specifically, a TRL of 2.

In July 2023, methanol became the fuel for 25 ships worldwide, and 127 terminals were successfully supplying ships with this fuel [119]. As previously mentioned, the technologies and procedures for using methanol as a marine fuel and for bunkering applications were established and are regulated by the IMO and classification societies. According to the report from the ABS [32], methanol-burning engines utilizing high-pressure diesel combustion processes have been made available by the manufacturers MAN and Wärtsilä. Moreover, methanol has been transported in chemical carriers for several decades and is also utilized by offshore support vessels (OSVs) and platform supply vessels (PSVs) for the offshore industry [32], facilitating its widespread adoption as a marine fuel. Due to these favourable factors and the potential for rapid integration into the marine fleet, methanol was estimated to possess high potential for widespread use in the short term. Nevertheless, the major source of methanol production is fossil-based (coal or natural gas) [124], presenting an obstacle to the widespread adoption of renewable methanol for maritime transport applications. As a result, the technological readiness level assigned to methanol as a marine fuel is TRL 7, indicating an advanced stage of technological development and readiness for practical implementation, yet renewable production still demands further expansion.

Ammonia currently benefits from an established supply chain network primarily catered to its use in the chemical industry [73] with efficient transportation via ships worldwide. The MAN dual-fuel engine, originally designed to operate with methanol and diesel, can also be adapted to use ammonia as an alternative fuel, provided certain modifications are made to the feed system's pressure [23]. As a result, the technologies, materials, and procedures necessary for its application are well-known within the industry. Nonetheless, further adaptation and development are required to utilize ammonia as a marine fuel [113]. The use of this fuel would face competition from the chemical sector and encounter challenges such as high toxicity and the technology's premature stage for integration into engines and fuel cells. Consequently, in order for ammonia to attain full commercial viability in the long term, further technological advancement is required, and as a result, the assumed TRL for ammonia is 5.

### 3.6. Summary of Results

In Table 4, a comparison between fuels is summarized by topics: energy density compared with HFO, bunkering readiness, material compatibility, storage tanks, engine feed, engine option, safety, and TRL.

**Table 4.** Summary of the comparison between fuels.

| Criteria | LNG | Biodiesel | SVO | HVO | HPO | Methanol | Ammonia |
|---|---|---|---|---|---|---|---|
| Energy density HFO/fuel | 1.91 | 1.21 | 1.19–1.08 | 1.15 | 1.19 | 2.47 | 2.81 |
| Bunkering readiness | Already worldwide established | Adaptation to biodiesel properties, narrow shaped tanks, constant cleaning | Procedures are similar to HFO bunkering | Procedures are similar to MDO bunkering | Urge of development all bunkering process | Under establishment, ventilation reinforcement | Ammonia bunkering is already performed in the chemical industry |
| Material compatibility | Aluminium and stainless steel | Stainless steel or zinc reinforcement | Stainless or mild steel if coated with zinc silicate | No changes are needed | Stainless steel | Stainless or austenitic manganese steel | Stainless steel |
| Storage tanks | Double-walled, cryogenic storage ($-162°$), 10 bar pressure, inert | Isolated from machinery | Isolated from machinery, coated with vegetable oil inert material | Constant maintenance to avoid water contamination | Isolated from machinery, coated with biomass oil inert material | Double-walled, detection system to leakages | Double-walled, isolated from machinery, pressure of 8.6 bar |
| Engine feed | Double-walled, Ventilation reinforcement, 10 bar feed pressure | Filtering, constant maintenance | Pre-heating (67 to 78 °C), filtering, constant maintenance | No changes are needed | Pre-heating, piping designed to not block solid particles, filtering | Double-walled, ventilation reinforcement, pressure of 10 bar | Double-walled, ventilation reinforcement |
| Engine option | Dual fuel | Diesel engine | Diesel engine | Diesel engine | Diesel engine | Dual fuel | Fuel cell (dual-fuel is also an option) |
| Safety | Cryogenic and flammable | Low temperature use restricted due to low pour point, low toxicity | Low toxicity | Low toxicity | Low toxicity | Highly toxic and flammable | Highly toxic and flammable |
| TRL | 9 | 7 | 5 | 5 | 2 | 7 | 5 |

## 4. Case Study

The Brazilian maritime sector has a fleet of approximately 2700 vessels [52] and more than 380 ports or terminals [125]. According to ANTAQ (Agência Nacional de Transportes Aquaviários) [52], long-haul navigation accounts for the highest cargo and travel movement, indicating the significant flow of Brazilian trade goods with foreign countries. Cabotage has some heavily travelled routes, such as Santos to Pecém, which is mainly focused on container transportation. However, this type of freight represents roughly one-third of the cargo and travel compared with deep-sea navigation. Concerning the energy transition of the maritime sector, the Brazilian Ministry of Mines and Energy (MME) initiated a program in 2012 aimed at the promotion of sustainable technologies applicable to all modes of transportation, particularly marine transport [126].

### 4.1. Main Port Profiles and Future Hubs

Brazilian port facilities exhibiting higher activity rates, as determined using the 2021 cargo movement data, namely, Ponta da Madeira, Santos, Tubarão, Angra dos Reis, São Sebastião, Paranaguá, Açu, Itaguaí, Itaqui, and Ilha da Guaíba [52], can be identified as primary hotspots for the transition of the Brazilian maritime transportation sector. Furthermore, ports and terminals with registered bunkering or movement of alternative fuels as cargo, meaning there is infrastructure in place to handle the loading or unloading of selected fuels, should also be accounted for. Finally, there are also ports that exhibit planned implementation of infrastructure dedicated to the bunkering of alternative fuels. Figure 3 summarizes Brazilian port information, classified according to the previously mentioned criteria.

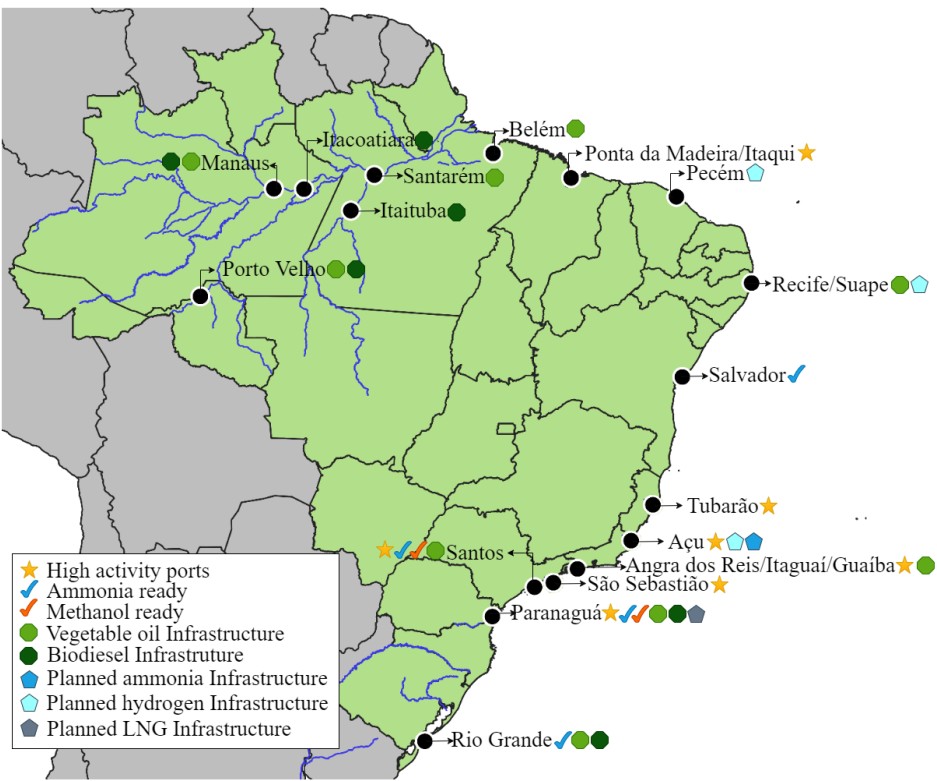

**Figure 3.** High-activity and alternative fuels available, available handling infrastructure, and planned ports and terminals.

Regarding bunkering, in July 2023, an agreement was finalized with ports and companies within the Brazilian maritime sector, with the primary objective of promoting the utilization of alternative fuels in ships [127]. Given the limited number of Brazilian ports equipped with the necessary infrastructure for bunkering non-conventional fuels, such

initiatives are of utmost importance in stimulating the transformation of Brazil's maritime infrastructure. As exposed in Figure 3, notably, the ports located in Santos, Rio Grande, Paranaguá and Salvador possess infrastructure for ammonia bunkering, whereas the facilities in Santos and Paranaguá are additionally equipped for methanol bunkering [119].

Figure 3 also shows ports and terminals that have the infrastructure to handle SVO and biodiesel. Since 2013, biodiesel has been transported by ships departing from various ports in Brazil, namely, Belém, Itacoatiara, Itaituba, Manaus, Paranaguá, Porto Velho, and Rio Grande [52]. Additionally, ANTAQ [52] completes the transportation of vegetable oils (specifically, palm and soybean) using specific Brazilian ports, including Barcarena, Belém, Manaus, Paranaguá, Porto Velho, Santos, Recife, Rio de Janeiro, Rio Grande, and Santarém. This indicates the existence of adequate infrastructure to handle the bunkering of vegetable oils and their derivatives at major ports throughout Brazil.

Furthermore, with regard to forthcoming adaptations, Paranaguá port has undertaken plans to construct infrastructure to facilitate LNG bunkering, with the projected beginning of operations in 2025 [128]. Simultaneously, the port is also actively investigating the implementation of a biodigester plant dedicated to the production of biomethane, which can be liquefied and turned into a green alternative to LNG [129]. Parallelly, in 2021, Pecém port created a proposal for the establishment of a hydrogen hub in its facilities [130]. This strategic move holds the potential to equip the ports with a dedicated infrastructure for the transportation and handling of hydrogen. As outlined earlier, hydrogen handling demands liquefaction and pressurization to optimize storage, along with precise conditions for loading and unloading operations [131]. Consequently, the procedures governing the handling of hydrogen closely mirror those already used for LNG and ammonia, rendering the port susceptible to the bunkering procedures of the aforementioned fuels.

The port of Açu also has plans to enable the bunkering of not only hydrogen but also ammonia. In partnership with the oil company Shell, the port authority is arranging the establishment of a facility dedicated to the production of the aforementioned fuels, along with the development of the necessary supply infrastructure [132]. Similarly, the port of Suape is also engaged in ongoing projects for the production of green hydrogen and ammonia [133].

The selected ports were also examined in terms of cargo movement, main products handled, and destinations. Table 5 displays their main compiled data.

**Table 5.** Total cargo movement (in millions of metric tonnes) in 2021, main products, and destinations departing from each analysed port.

| Port | Cargo Movement ($10^6$ Metric-Ton) | Main Products | Main Destinations |
|---|---|---|---|
| Açu | 39.0 | Oil and derivatives, containers, cooper, iron and steel | Suape, Madre de Deus, Santos, Rio de Janeiro, Vitória |
| Angra dos Reis | 29.3 | Iron and steel, oil and derivatives | Alexandria (Egipt), Mersin (Turkey), Kabil (Indonesia), Qingdao (China), Aratu |
| Belém | 2.6 | Containers, oil and derivatives, corn, general cargo | Manaus, Barcarena, Fortaleza, Madre de Deus, Santarém |
| Guaíba | 26.3 | Iron ore, wood, cellulose pulp | Rio de Janeiro, Rio Grande, Port Talbot (Wales), Ijmuiden and Rotterdam (the Netherlands) |
| Itacoatiara | 7.0 | Soy, soy oil, ethanol, fossil fuels, oil and derivatives | Fortaleza, Manaus, Itaqui |
| Itaguaí | 46.9 | Containers | Santos, Imbituba, Suape, Callao (Peru), Rotterdam (the Netherlands) |
| Itaituba | 6.1 | Oil and derivatives, corn, soy | Belém, Manaus, Porto Velho, Santarém, Santana |
| Itaqui | 20.3 | Oil and derivatives, containers, ethanol, chemical products | Belém, Aratu, Fortaleza, Santos, Suape |
| Manaus | 6.0 | Oil and derivatives, containers, general cargo | Belém, Fortaleza, Santos, Suape, Itacoatiara |

**Table 5.** *Cont.*

| Port | Cargo Movement (10$^6$ Metric-Ton) | Main Products | Main Destinations |
|---|---|---|---|
| Paranaguá | 32.6 | Containers, oil and derivatives, chemical products, wheat | Belém, Fortaleza, Santos, Suape, Itaguaí |
| Pecém | 10.4 | Containers, iron and steel, oil and derivatives, manganese | Los Angeles (USA), Manaus, Cubatão, Brownsville (USA), Santos |
| Ponta da Madeira | 186.6 | Iron ore | Qingdao (China), Labuan (Malaysia), Kwangyang (Korea), Sohrar (Oman), Pecém |
| Porto Velho | 14.2 | Soy, corn, containers, general cargo | Santarém, Itacoatiara, Belém, Long Beach (USA), Montoir De Bretagne (France) |
| Recife | 0.3 | Sugar, salt, oil and derivatives, fossil fuels | Dubai (UAE), Fernando de Noronha, Baltimore (USA), Barra Do Riacho, Douala (Cameroon) |
| Rio Grande | 20.0 | Soy, containers, wood, fertilizers | Tanger (Morocco), Pecém, Antwerpen (Belgiun), Porto Alegre, Dafeng (China) |
| SãoSebastião | 12.6 | Oil and derivatives, sugar | Singapore, Qingdao (China), Manaus, Itaqui, Itacoatiara |
| Salvador | 4.5 | Oil and derivatives, cellulose pulp, containers | Vila do Conde, Belém, São Sebastião, Changshu (China), Santos |
| Santarém | 6.5 | Oil and derivatives, soy, corn, fertilizers | Itaituba, Algete and Barcelona (Spain), Belém, Rotterdam (the Netherlands) |
| Santos | 99.1 | Soy, oil and derivatives, soy oil, containers | Anshan, Koh Sichang (China), Bandar Khomeini (Iran), Singapore, São Sebastião |
| Suape | 11.8 | Oil and derivatives, containers, sugar, ethanol | Singapore, Manaus, Fortaleza, Itaqui, Santos |
| Tubarão | 62.7 | Iron ore, soy | Tangshan, Qingdao and Rizhao (China), Labuan (Malaysia), Rio de Janeiro |

Data from ANTAQ [52].

One important outlook of the analysis of main Brazilian ports is that shipping is focused on bulk and container products. Routes are diverse, yet most of the cargo movements are concentrated in international destinations, confirming the importance of long-haul navigation to Brazil's economy. China is the busiest destination for Brazilian exports, mainly due to iron ore, soy, corn, oil, and containers [52]. Another output is the high activity in the Brazilian north region, mostly in the Legal Amazon Area. Ports such as Ponta da Madeira, Manaus, Belém, Porto Velho, and Santarém heavily contribute to local shipping.

Considering cargo movement and the potential conversion of ports for the bunkering of alternative fuels, it can be concluded that ports characterized by high cargo movement—herein presumed to be ports sustaining an annual cargo movement greater than 10 million tonnes—alongside a diverse product flow, encompassing a minimum of four distinct products categories, and consequently having a varied array of types of ships docked, are more acceptable for implementation as multi-fuel hubs. The ports satisfying these criteria, as listed in Table 5, encompass Açu, Itaqui, Paranaguá, Porto Velho, Rio Grande, Santos, and Suape.

Additionally, ports that envision the integration of infrastructure designed to enable the provision of two or more alternative fuel bunkering exhibit heightened precedence in relation to the establishment of multi-fuel hubs. Ports that have handled any of the analysed fuels as cargo also meet this criterion. Specifically, as Figure 3show, these ports are Açu, Manaus, Paranaguá, Porto Velho, Santos, Suape, and Rio Grande.

Taking into account the two abovementioned criteria, our analysis delineates the following ports as possessing the potential to serve as a multi-fuel hub: Açu, Paranaguá, Porto Velho, Rio Grande, Santos, and Suape.

Conversely, ports such as Ponta da Madeira, Itaguaí, and Tubarão, distinguished by substantial cargo movement, although with a concentrated product range, are assessed to be prone to experiencing a more restricted bunkering of alternative fuels. In other words, these ports are better suited to the bunkering of a particular alternative fuel, considering

factors such as the final destinations of the product's fuel availability, and even the local production disposal of alternative fuels.

### 4.2. Fleet and Cargo Profile: Challenges and Progress in Conversion to Alternative Fuel Use

In 2023, the Brazilian ship fleet recorded an average age of approximately 19.5 years. Support vessels, despite being smaller, stand out due to their significant quantity, representing 90% of the fleet. Port support vessels account for 73% of this total, while maritime support vessels represent 27% [52]. Among the ships with the highest gross tonnage, bulk carriers and container ships are highlighted. Based on ANTAQ [52], Table 6 displays the products transported, age, and average deadweight tonnage (DWT), along with the number of ships, for the types of vessels with the highest average DWT in the Brazilian fleet.

**Table 6.** Products transported, average age, deadweight tonnage, and number of ships of the main Brazilian ship types.

| Ship Type | Products Transported | Average Age (2023) | Average DWT | Fleet Size |
|---|---|---|---|---|
| Tanker | Crude oil and derivatives | 10 | 89,054 | 54 |
| Bulk | Dry bulk | 15 | 57,007 | 21 |
| Container | Container | 13 | 45,009 | 33 |
| Chemical tanker | Chemical products | 18 | 26,234 | 8 |
| Pipe laying support vessel (PLSV) | Offshore pipes | 9 | 10,661 | 8 |
| Subsea equipment support vessel | Subsea equipment | 15 | 7570 | 2 |
| LPG tanker | Liquefied petroleum gas | 11 | 5481 | 8 |
| Liquefied gas tanker | Liquefied gases | 13 | 5455 | 11 |

[52].

Given that the typical lifespan of a ship is 30 years [50], it can be concluded that the highlighted types of vessels exhibit a residual lifespan of no less than 12 years, a scenario particularly applicable to the chemical tanker fleet. Therefore, replacement of the existing fleet due to the end of its lifetime remains an impractical course of action for a short period. In this regard, a priority arises to optimize the ship retrofits required for the adoption of alternative fuels.

LPG and liquefied gas tankers are notably suited to embrace the utilization of liquefied and pressurized fuels, namely, LNG, ammonia, and methanol. This advantage stems from the existing infrastructure designed for the storage and management of these fuels, which leads to a simplified conversion than other vessels.

Chemical tankers are also more suitable for ammonia and methanol. These fuels are flammable, demanding ships to be meticulously constructed and operated with intensified attention to potential incidents concerning the cargo [134]. This condition particularly applies to chemical ships, easing the adaptation to the use of the aforesaid fuels.

Tanker ships also exhibit a notable advantage in terms of adaptability due to their operation with fuel as cargo. However, changes in the entire infrastructure, encompassing storage tanks, fuel feeding and engines, are imperative. Given their intrinsic lack of operational experience with liquefaction and extreme pressurization, these vessels are better suited to undergo conversion for the utilization of other fuels, preferably having higher readiness levels, such as biodiesel, SVO and HVO. An analogous circumstance applies to the remaining selected types of vessels, given their inherent limitation of lacking experience in the handling of fuel as cargo.

Concerning the current stage of fuel usage, in 2022, Bunker One, a Danish bunkering company actively engaged in operations along the Brazilian coast, entered into a collaborative partnership with the Federal University of Rio Grande do Norte to conduct experimental trials on a fuel blend composed of HFO and 7% $v/v$ biodiesel. These trials are specifically focused on tugboats operating within the area of the Port of Rio de Janeiro, with the aim of gathering valuable data on the performance and suitability of this mixture

in the maritime context [135]. Petrobras has undertaken the implementation of a fuel blend consisting of 90% HFO and 10% biodiesel in an LPG tanker, with the primary objective of conducting a comprehensive analysis of its performance characteristics and identifying any potential logistics challenges that may arise. The dedicated Research Laboratories at Petrobras conducted testing and assessment of this fuel mixture in January 2023, observing that its integration necessitates no modifications to the existing maritime infrastructure [136]. In July 2023, the company made an announcement regarding its plans to conduct additional tests on vessels using a blend of 24% *v/v* of biodiesel [137]. Additionally, the company is actively investing in and establishing the development of large-scale production of HVO within its refineries [138].

As previously mentioned, companies linked to the maritime and energy sectors have taken the lead in the effort to introduce alternative fuels into vessels. Apart from these companies, governmental and regulatory bodies must be prepared to assume a pivotal role in facilitating the transition of the maritime sector [7]. Their contribution encompasses measures targeted not only at facilitating fuel production but also at proposing the conversion of marine fleet and port infrastructure. The actions of governments, such as those in Norway, range from setting more ambitious targets relative to those defined by IMO, directing mandatory percentages of biofuels within maritime fuel blends, to instituting fiscal incentives for enterprises that champion the utilization of alternative fuels [139], present examples that Brazil could consider to follow.

### 4.3. Thermal Stability of Fuels in the Main Routes

In terms of the thermal stability of the selected fuels, as highlighted in Sections 3.1 and 3.6, biodiesel exhibits a low pour point compared to traditional marine fuels and other alternative fuels. This particular property restricts its widespread usage in regions characterized by low temperatures or during cold seasons [45]. Given the routes departing from the main Brazilian ports, displayed in Table 4, and global historical average temperatures across various regions [140], it can be concluded that international routes transiting through South Africa, Europe, the United States, and North Asia demand the use of distinct fuels from biodiesel during periods of low temperature.

### 4.4. Fleet Profile: Loss of Cargo Space

Shipping companies, particularly those specializing in long-haul navigation, are continuously in search of strategies to optimize the allocation of cargo freight, aiming to maximize its utilization during a voyage. This pursuit explains the quest for achieving economies of scale in bulk shipping [141], whose vessels have progressively larger cargo capacities. For instance, standard dry bulk carriers have reached a capacity of 400,000 DWT with the deployment of Valemax vessels, the regular ships for the Ponta da Madeira to Qingdao iron ore route [142]. As clarified in Section 3, the adoption of alternative fuels brings a consequential requirement for increased storage tank volume due to the relatively lower energy density in contrast to conventional fuels. This decrease in space availability, particularly seen in the cases of LNG, ammonia, and methanol, is set to decrease the allocation of cargo space [46]. Given the substantial reliance on bulk shipping in the Brazilian context, this loss of cargo space emerges as a considerable barrier to the effective use of alternative fuels. In response to this challenge, Lindstad et al. [143] proposed some initiatives aimed at mitigating the loss of cargo space, including the increase in maximum draught and length of vessels. In the short term, however, this loss of space tends to be solved with more ships [144].

## 5. Conclusions

This study reviewed and summarized the major changes required for ports and long-distance large cargo ships to store, feed, and use alternative fuels. Considering the focus on fuel usage, this work did not encompass aspects related to the production chain, such as feedstock diversity. Therefore, no distinctions were made between fossil, bio, and e-fuels.

The handling, bunkering, and usage of alternative fuels must deal with: (i) the low energy density of fuels compared with HFO, particularly, LNG, ammonia, and methanol, leading to a loss in cargo space; (ii) the need for liquefaction (LNG) and/or pressurization (ammonia and methanol) of fuels to optimize storage or enable proper fuel feeding; (iii) the use of different materials such as stainless and mild steel in storage tanks and fuel feeding systems; (iv) the requirement for double-walled storage tanks and fuel feeding systems, as observed in the cases of LNG, ammonia, and methanol; (v) the need for enhanced precautions to prevent water contamination, particularly to biofuels usage; (vi) the high toxicity of fuels, notably, ammonia and methanol, which require extra ventilation inside ships; (vii) thermal stability issues impacting biodiesel utilization, particularly in extreme low temperatures; and (ix) modifications in engine fuel feeding and ignition (biofuels), adjustments for dual-fuel (LNG and methanol), and substitution for fuel cells (ammonia).

It is worth noting that although economic factors are not discussed in this study, they represent a challenge for the marine and academic communities, as evidenced by the research conducted by DNV GL [57], Xing et al. [40], UMAS [145], Bilgili [46], and Carvalho et al. [49]. Alternative fuels are still costlier than fossil fuels due to their more expensive production and capital and the operational cost of vessels, especially ammonia and hydrogen [46]. Economic competitiveness will be unreachable without actions from stakeholders to enhance alternative fuel usage, such as incentives, national and regional policies, and carbon pricing [9].

While the demand for alternative fuels is increasing, further advancement is necessary to significantly broaden the array of options. While fuels like LNG and methanol are already in use on specific vessels, fuels like HPO and SVO are still in the experimental stage. This posed challenges when reviewing technical and scientific literature related to these emerging fuel options.

The conducted case study underscored the feasibility of single or multi-fuel bunkering within the main Brazilian ports by indicating the main products and routes and the prospective development of alternative bunkering infrastructure within each port studied. Ports such as Açu, Paranaguá, Porto Velho, Rio Grande, Santos, and Suape exhibit potential for accommodating multi-fuel bunkering, while Ponta da Madeira, Itaguaí, and Tubarão tend to accommodate single-fuel bunkering.

Concerning the Brazilian fleet, given the limited number of alternative fuel trials within the country, the analysis int this study was conducted by evaluating vessel types requiring fewer adaptations for the utilization of alternative fuels. Due to the operational characteristics of ships, LPG and liquefied gas tankers are leading the way in terms of conversion to utilize fuels like LNG, ammonia, and methanol. A similar trend is observed for chemical vessels, which are more suitable for conversion to ammonia and methanol use, as well as tanker ships, which hold potential for the use of fuels such as biodiesel, SVO, and HVO. In the pursuit of establishing a fleet powered by alternative fuels, stakeholders may adopt diverse strategies, including the establishment of more ambitious targets, mandatory incorporation of biofuels in blends, and fiscal incentives promoting the integration of alternative fuels in their fleets. The analysis of these different strategies should be deepened in further studies. Further studies could also widen our analysis to other types of ships, for example, those that are more appropriate for hydrogen and electric batteries, such as ferryboats, offshore support vessels, etc. Additionally, the implementation of alternative fuel bunkering in a specific port can be a factor in reducing port congestion. Shipping companies struggle to avoid queuing in port areas and search for alternatives to avoid it, such as adapting shipping routes and destination ports in order to diminish cost losses [146]. The impact of a wide network of alternative fuel bunkering ports can be evaluated in future studies targeting not only port congestion reduction but also energy inflation reduction, which are intrinsically related [98].

**Author Contributions:** Conceptualization, H.W., C.R.P.B. and A.S.; methodology, H.W.; formal analysis, H.W., E.M.-C. and A.S.; investigation, H.W.; writing—original draft preparation, H.W.; writing—review and editing, E.M.-C. and A.S. All authors have read and agreed to the published version of the manuscript.

**Funding:** This research received no external funding.

**Institutional Review Board Statement:** Not applicable.

**Informed Consent Statement:** Not applicable.

**Data Availability Statement:** Not applicable.

**Acknowledgments:** We thank the Brazilian agencies CNPq and CAPES for their support on the early stages of this study.

**Conflicts of Interest:** The authors declare no conflict of interest.

## Notes

[1] Net zero emissions are achieved when human caused GHG emissions are balanced globally by human induced removals of $CO_2$ on a global scale during a defined period [10].

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
