# Peer review of "Evaluating the Readiness of Ships and Ports to Bunker and Use Alternative Fuels: A Case Study from Brazil"

_jmse, doi:10.3390/jmse11101856_

Round 1
Reviewer 1 Report
Comments and Suggestions for Authors
Please see attachment.

Author Response
Dear reviewer,
We sincerely appreciate your feedback, and we have incorporated your comments into the revised manuscript. Here are our responses to your feedback.
- I would suggest to the authors to include a paragraph that would briefly describe the economic implications of the different alternative fuels described. The financing perspective between the current fossil fuels and the alternative ones is am major issue both practitioners but also for the academic community.
We express our gratitude to the reviewer for their suggestion, and in response, we have added the following paragraph as an addition to our manuscript:
“It is worth noting that, although economic factors are not discussed by this study, they represent a challenge for the marine and academic communities, as evidenced by the research conducted by DNV GL [1], Xing et al. [2], UMAS [3], Bilgili [4] and Carvalho et al. [5]. Alternative fuels are still costlier than fossil fuels due to their still more expensive production and capital and operational costs of vessels, especially in the case of ammonia and hydrogen [4]. Economic competitiveness will be unreachable without actions from stakeholders to enhance alternative fuels usage, such as incentives, national and regional policies, and carbon pricing [6].”
- I would suggest to the authors to include a paragraph that states implications of port congestion in the modern world for example during the Covid-19 pandemic or the Ukrainian – Russian war. The authors can discuss in this paragraph if alternative fuels can also diminish port congestion which has been proven to increase energy inflation. The authors may find the following articles useful for their research:
- Stopford, M., 2008. Maritime economics 3e. Routledge.
- Karamperidis, Stavros and Michail, Nektarios and Melas, Konstantinos, LNG Carriers Discharge Waiting Time and Energy Inflation (May 31, 2023). Available at http://dx.doi.org/10.2139/ssrn.4464776
- Michail, N.A. and Melas, K.D., 2020. Shipping markets in turmoil: Na analysis of the Covid-19 outbreak and its implications. Transportation Research Interdisciplinary Perspectives, 7, p.100178.
We thank you for your comment. We added the following text to the article and cited alternative fuels as an option to reduce port congestion as a possible future study:
“Additionally, implementation of alternative fuels bunkering in a specific port can be a factor to reduce port congestion. Shipping companies struggles to avoid queuing in port areas, searching alternatives to avoid it, such as adapting shipping routes and destination ports in order to diminish cost losses [7]. The impact of a wide network of alternative fuels bunkering ports can be evaluated in future studies targeting not only port congestion reduction, but also energy inflation reduction, which are intrinsically related [8].”
References:
- DNV GL Comparison of Alternative Marine Fuels. 2019.
- Xing, H.; Stuart, C.; Spence, S.; Chen, H. Alternative Fuel Options for Low Carbon Maritime Transportation: Pathways to 2050. J. Clean. Prod. 2021, 297, 126651, doi:10.1016/j.jclepro.2021.126651.
- UMAS Techno-Economic Assessment of Zero-Carbon Fuels. 2020.
- Bilgili, L. A Systematic Review on the Acceptance of Alternative Marine Fuels. Renew. Sustain. Energy Rev. 2023, 182, 113367, doi:10.1016/j.rser.2023.113367.
- Carvalho, F.; Müller-Casseres, E.; Poggio, M.; Nogueira, T.; Fonte, C.; Wei, H.K.; Portugal-Pereira, J.; Rochedo, P.R.R.; Szklo, A.; Schaeffer, R. Prospects for Carbon-Neutral Maritime Fuels Production in Brazil. J. Clean. Prod. 2021, 326, doi:10.1016/j.jclepro.2021.129385.
- Serra, P.; Fancello, G. Towards the IMO’s GHG Goals: A Critical Overview of the Perspectives and Challenges of the Main Options for Decarbonizing International Shipping. Sustain. 2020, 12, doi:10.3390/su12083220.
- Meng, L.; Ge, H.; Wang, X.; Yan, W.; Han, C. Optimization of Ship Routing and Allocation in a Container Transport Network Considering Port Congestion: A Variational Inequality Model. Ocean Coast. Manag. 2023, 244, doi:10.1016/j.ocecoaman.2023.106798.
- Karamperidis, S.; Michail, N.; Melas, K. LNG Carriers Discharge Waiting Time and Energy Inflation. SSRN Electron. J. 2023, 1–10, doi:10.2139/ssrn.4464776.

Reviewer 2 Report
Comments and Suggestions for Authors
The manuscript is quite engaging, although I have some significant concerns outlined below:
1. The authors should delve into the potential contributions of candidate alternative fuels to decarbonising the shipping industry, with a focus on Life Cycle Greenhouse Gas Emission Assessment. It's important to consider that methanol and ammonia can be produced from fossil fuels, which is meaningless for decarbonising shipping industry. You may refer to relevant publications in JMSE.
2. While fossil LNG is seen as a transitional fuel, Bio-LNG and E-LNG have the potential to serve as drop-in fuels. Several European ports already offer Bio-LNG as a drop-in fuel. Why don’t you consider Bio-LNG as a partial drop-in fuel?
3. Hydrogen has a significant role to play in decarbonising the shipping industry, with several MW-level hydrogen fuel cell-powered vessels already being ordered. Why hasn't hydrogen been considered as a candidate alternative fuel for shipping in your manuscript? Your explanation is insufficient.
4. Some of the candidate alternative fuels, such as LNG, ammonia, and methanol, come with safety risks. It is advisable for the authors to discuss these risks associated with the use and bunkering of these fuels, especially emphasising LNG and ammonia bunkering risks.
5. In terms of energy conversion, the authors have emphasised internal combustion engines, but it's worth mentioning that fuel cells could also capture a share of the market. Consider discussing fuel cells more extensively.
6. Table 3 would benefit from the inclusion of minimum ignition energy and self-ignition temperature data.
7. Line 313 on Page 9 states, "Methanol,…,requires pressurization," which is incorrect.
8. In section 3.2, the discussion on bunkering safety seems insufficient, particularly for LNG and ammonia. I recommend adding a more comprehensive discussion on LNG bunkering safety and ammonia bunkering safety. You may refer to the provided references as below for ammonia bunkering safety.
https://doi.org/10.1002/prs.12326
https://doi.org/10.1016/j.jhazmat.2023.131281
9. Line 348 of Page 9, it mentions, "All fuelling methods applicable to LNG can be employed for methanol as well." This should be ammonia instead of methanol.
10. In section 3.3, while the manuscript covers storage for LNG and ammonia, it lacks discussion on fuel supply systems (fuel handling systems or fuel feeding systems), which are crucial due to the low temperature characteristics of these fuels. Please consider adding this discussion.
11. Line 486 on Page 12 states, "with a particular emphasis on Tankers." This sentence is unclear. LNG is used as fuel in various types of vessels, not just tankers. Please clarify your intended meaning.
12. On Line 554 of Page 13, the manuscript mentions TRL 8 for methanol bunkering, which may not be appropriate. The challenge lies in supplying green methanol for the shipping industry, as existing methanol supplies are fossil fuel-based. Please revise accordingly.
13. In Table 4, under "Engine option for ammonia," consider including "Dual fuel" and "fuel cell." MAN and Wärtsilä are developing ammonia engines, which may become more popular than ammonia fuel cells. Additionally, under "Safety for LNG," include "Cryogenic" and "Flammable."
14. The manuscript could benefit from some shortening to enhance readability.
I hope these revisions and suggestions help improve the manuscript.
05/09/2023
Author Response
Dear reviewer,
We sincerely appreciate your feedback, and we have incorporated your comments into the revised manuscript. Here are our responses to your feedback.
- The authors should delve into the potential contributions of candidate alternative fuels to decarbonising the shipping industry, with a focus on Life Cycle Greenhouse Gas Emission Assessment. It's important to consider that methanol and ammonia can be produced from fossil fuels, which is meaningless for decarbonising shipping industry. You may refer to relevant publications in JMSE.
We thank the reviewer for her/his comment that helped us to highlight the major focus of our study. Actually, we added text to the main manuscript in order to explain our aims:
“…Some of the highlighted fuels can have their production based on both fossil and sustainable sources. For instance, LNG, methanol, ammonia and hydrogen can be produced from fossil, biomass, or via electrolysis and Carbon Capture and Storage (CCS) or Direct Air Capture (DAC), known as e-fuels [1]. Both bio and e-fuel alternatives possess the potential to reduce GHG emissions when compared to fossil-based fuels [2]. Given that the primary goal of this analysis is to assess the compatibility of alternative fuel’s handling, storage and usage, the discussion done does not encompass the evaluation of the GHG emissions of these fuels from their production to their consumption. This has being done by several works, such as Muller-Casseres et al. [3] and Brynolf et al. [4].”
“This study reviewed and summarized the major changes required for ports and long-distance large cargo ships to store, feed and use alternative fuels. Considering the focus on fuel usage, this work did not encompass aspects related to the production chain, such as feedstock diversity. Therefore, no distinctions were made between fossil, bio, and e-fuels.”
“As mentioned before, it is worth noting here that ammonia, LNG and methanol can be produced from fossil, bio and synthetic feedstocks. Our focus here is not on their production but on their handling and usage…”
- While fossil LNG is seen as a transitional fuel, Bio-LNG and E-LNG have the potential to serve as drop-in fuels. Several European ports already offer Bio-LNG as a drop-in fuel. Why don’t you consider Bio-LNG as a partial drop-in fuel?
As stated in the last answer, we tried to review the paper to highlight that methane and methanol can have renewable or fossil feedstocks. We thank the reviewer for her/his comment on this. We have also better explained the meaning of a fuel being partially drop-in. For instance, we defined that:
“Some of the non-drop in fuels have already an established infrastructure in several ports (case of the tradeable ammonia and methanol) and have a relevant usage record by dual engines (LNG and in a less extent methanol). However, the categorization done here considered that more than 95% of large ships are still based on diesel engines and the ports associated with their routes are mostly single hubs to store and bunker petroleum derived fuels for them [5].”
- Hydrogen has a significant role to play in decarbonising the shipping industry, with several MW-level hydrogen fuel cell-powered vessels already being ordered. Why hasn't hydrogen been considered as a candidate alternative fuel for shipping in your manuscript? Your explanation is insufficient.
Once again we would like to thank the reviewer since her/his comment helped us to clearly explain our focus. Actually, we agree that hydrogen powered vessels will be increasingly built. The same for hybrid vessels. We tried to say that in the paper, by stressing that:
“Recently, there has been a substantial number of studies delving into the subject of the production and consumption of biofuels [6–10], hydrogen and ammonia [11–15], liquefied natural gas (LNG) [16–18], and methanol [19–22] for shipping. While a significant share of these studies focuses on the technical aspects of production [3,4,23–27], emissions mitigation [2,28–30], and their consumption in marine engines [31–33], few have given due attention to the necessary adaptations required in ships and ports to the operation of these alternative fuels.”
However, as we tried no to stress, “the primary aim of this study is to assess the current progress of adjusting ships and ports to effectively use selected alternative fuels, with a particular emphasis on their applicability to long-haul cargo shipping, mostly characterized by large vessels, which significantly contributes to the sector’s overall energy demand and GHG emissions [34].”
“In addition, since this study addresses the long distance freight transportation based on large vessels, energy carriers and storage options, such as hydrogen and batteries, are not evaluated given their low suitability for deep-sea large ships, as showed by Gray et al. [35] and Xing et al. [26]. Indeed, these alternatives leads to a substantial spatial allocation loss in comparison to conventional fuels, making them impractical for long-distance shipping [36].”
As we wrote in the revised conclusion section, “Further studies could also widen our analysis to other types of ships, for example more appropriated to hydrogen and electric batteries, such as ferryboats, offshore support vessels, etc.”
- Some of the candidate alternative fuels, such as LNG, ammonia, and methanol, come with safety risks. It is advisable for the authors to discuss these risks associated with the use and bunkering of these fuels, especially emphasising LNG and ammonia bunkering risks.
Thanks for the comment. We have emphasized this important property in the revised manuscript.
“Methanol [37] and ammonia [38] are widely employed as feedstocks in the chemical industry. Given their high toxicity, it is essential to implement safety measures to prevent leaks and human exposure to these substances, such as gas detectors. As stated by Kay et al [39], ammonia leakage not only in air, but also in sea, can lead to critical damages, and lethality can be greatly reduced if release duration is shortened: the authors found that a 30 seconds leakage is 70% less lethal than a 60 seconds leakage. Safety measures must be targeted in mitigate toxicity, especially at potential sources of leakage such as inlet and outlet manifolds for hose connection.
For ammonia, a refrigerated storage is preferable due to better effectiveness in reducing operational risk [39].
Flammability of ammonia [40], methanol [19], and LNG requires safety protocols to prevent the risk of leaks and spills, particularly in areas where ignition sources are present [41].”
- In terms of energy conversion, the authors have emphasised internal combustion engines, but it's worth mentioning that fuel cells could also capture a share of the market. Consider discussing fuel cells more extensively.
Thank you again. We revised the manuscript and tables in the results section to address the comment. For instance, we added this text:
“Xing et al. [36] stated that recent research and demonstration projects have validated the technical feasibility of fuel cell for maritime applications regarding power capacity, safety, durability and operational terms. These developments contribute to promote the adoption of fuel cell in vessel fleet in the future. However, it is important to note that despite these advancements, commercial viability remains a challenge [32], and their suitability for long-haul shipping is still limited [36].”
- Table 3 would benefit from the inclusion of minimum ignition energy and self-ignition temperature data.
Many thanks for the comment.
Due to lack of data for minimum ignition energy of fuels such as SVO and pyrolysis oil, data was not included. Self-ignition temperature was added.
- Line 313 on Page 9 states, "Methanol,…,requires pressurization," which is incorrect.
Thanks for showing us this tip. The sentence was removed from the manuscript.
- In section 3.2, the discussion on bunkering safety seems insufficient, particularly for LNG and ammonia. I recommend adding a more comprehensive discussion on LNG bunkering safety and ammonia bunkering safety. You may refer to the provided references as below for ammonia bunkering safety.
Thanks. We added more discussion on this subject. Please see for example the text below.
“Aneziris et al. [41] asserted that the utilization of low temperature pipelines, loading arms, and hoses is mandatory for LNG bunkering. Furthermore, the authors also highlight that it is essential to acknowledge that extremely low LNG temperatures may pose a significant hazard, impacting not only the structural integrity of materials employed by leading to potential cracks but also the safety of individuals in proximity to the LNG due to the risk of frostbite.
Those adjustments are demanded due to toxicity and flammability issues of ammonia, as previously addressed in Section 3.1 and extensively examined by Fan and Enshaei [40] and by Kay et al. [39]. The latter study also recommended the use of multiple hoses with lower flow rates instead of a single hose with higher flow rate should be employed, thereby resulting in a reduction of bunkering time and increased safety conditions.”
- Line 348 of Page 9, it mentions, "All fuelling methods applicable to LNG can be employed for methanol as well." This should be ammonia instead of methanol.
Thanks. We fixed this mistake.
- In section 3.3, while the manuscript covers storage for LNG and ammonia, it lacks discussion on fuel supply systems (fuel handling systems or fuel feeding systems), which are crucial due to the low temperature characteristics of these fuels. Please consider adding this discussion.
Thanks. We fully agree that this important subject was not suitably addressed in the first version of the manuscript. Now we tried to fix this mistake.
“Regarding engine fuel supply, in order to enhance LNG safety procedures, ABS [42] recommend the utilization gas detection systems for instant shutdown, double-wall piping with at least 30 air changes per hour, a maximum 10 bar pressure limit, nitrogen-based inertization for emergencies, and independent pumps and compressors from other circuits.
Parte superior do formulário
Additionally, with respect to fuel feeding, it is recommended to avoid corrosive materials such as copper, high-nickel alloys, and plastic. To prevent corrosion, it is advisable to use Teflon in engine seals instead of rubber and plastic [13]. A system for emergency ventilation must be installed and operated in accordance with either of the following principles: the reduction of ammonia concentration to below 10 ppm through dilution or the capture of excessive ammonia [43]. In order to reduce the potential risks associated with ammonia leakage in engine room, it is advisable to install both tank and feed system on the deck, coupled with its connection to the engine through dual-walled piping. Another alternative is to place the feed system and tanks within the engine room if an airlock system to prevent ammonia dispersion on-site is installed [44]. DNV GL [45] suggested a mandatory provision of secondary enclosures for all fuel piping to securely contain any potential leaks. Furthermore, an arrangement involving the infusion of nitrogen into the secondary enclosure, coupled with ongoing pressure monitoring, can also be an alternative solution to ensure safety.”
- Line 486 on Page 12 states, "with a particular emphasis on Tankers." This sentence is unclear. LNG is used as fuel in various types of vessels, not just tankers. Please clarify your intended meaning.
Thanks. We opted to withdraw the sentence from the manuscript.
- On Line 554 of Page 13, the manuscript mentions TRL 8 for methanol bunkering, which may not be appropriate. The challenge lies in supplying green methanol for the shipping industry, as existing methanol supplies are fossil fuel-based. Please revise accordingly.
Thanks for the comment. We changed the analysis accordingly.
“The major source of methanol production is fossil based (coal or natural gas) [46], presenting an obstacle to the widespread adoption of renewable methanol for maritime transport applications. As a result, the technological readiness level assigned to methanol as a marine fuel is TRL 7, indicating an advanced stage of technological development and readiness for practical implementation of methanol, yet renewable production still demands further expansion.”
- In Table 4, under "Engine option for ammonia," consider including "Dual fuel" and "fuel cell." MAN and Wärtsilä are developing ammonia engines, which may become more popular than ammonia fuel cells. Additionally, under "Safety for LNG," include "Cryogenic" and "Flammable."
The requested changes have been incorporated into the manuscript.
- The manuscript could benefit from some shortening to enhance readability.
We tried to shorten the manuscript, although we had to add some new text to answer the reviewer’s queries.
References
- Carvalho, F.; Müller-Casseres, E.; Poggio, M.; Nogueira, T.; Fonte, C.; Wei, H.K.; Portugal-Pereira, J.; Rochedo, P.R.R.; Szklo, A.; Schaeffer, R. Prospects for Carbon-Neutral Maritime Fuels Production in Brazil. J. Clean. Prod. 2021, 326, doi:10.1016/j.jclepro.2021.129385.
- Gilbert, P.; Walsh, C.; Traut, M.; Kesieme, U.; Pazouki, K.; Murphy, A. Assessment of Full Life-Cycle Air Emissions of Alternative Shipping Fuels. J. Clean. Prod. 2018, 172, 855–866, doi:10.1016/j.jclepro.2017.10.165.
- Müller-Casseres, E.; Carvalho, F.; Nogueira, T.; Fonte, C.; Império, M.; Poggio, M.; Wei, H.K.; Portugal-Pereira, J.; Rochedo, P.R.R.; Szklo, A.; et al. Production of Alternative Marine Fuels in Brazil: An Integrated Assessment Perspective. Energy 2021, 219, doi:10.1016/j.energy.2020.119444.
- Brynolf, S.; Fridell, E.; Andersson, K. Environmental Assessment of Marine Fuels: Liquefied Natural Gas, Liquefied Biogas, Methanol and Bio-Methanol. J. Clean. Prod. 2014, 74, 86–95, doi:10.1016/j.jclepro.2014.03.052.
- Smith, T.W.P.; Jalkanen, J.P.; Anderson, B.A.; Corbett, J.J.; Faber, J.; Hanayama, S.; O’Keeffe, E.; Parker, S.; Johansson, L.; Aldous, L.; et al. Third IMO Greenhouse Gas Study 2014; 2014;
- ABS Biofuels as Marine Fuel; 2021;
- Florentinus, A.; Hamelinck, C.; Bos, A. van den; Winkel, R.; Maarten, C. Potential of Biofuels for Shipping; 2012;
- IEA Biofuels for the Marine Shipping Sector; 2017;
- Winnes, H.; Fridell, E.; Hansson, J. Biofuels for Low Carbon Shipping. 2019.
- Laursen, R.; Barcarolo, D.; Patel, H.; Dowling, M.; Penfold, M.; Faber, J.; Király, J.; van der Ven, R.; Pang, E.; van Grinsven, A. Update on Potential of Biofuels in Shipping; Lisbon, 2022;
- ABS Ammonia As Marine Fuel. NH3 Fuel Conf. 2020.
- ABS Hydrogen as Marine Fuel. Sustain. Whitepaper 2021.
- Ash, N.; Scarbrough, T. Sailing on Solar. Could Green Ammonia Decarbonise International Shipping?; 2019;
- Earl, T.; Ambel, C.C.; Hemmings, B.; Gilliam, L.; Abbasov, F.; Officer, S. Roadmap to Decarbonising European Shipping. Transp. Environ. 2018, 22.
- Hansson, J.; Brynolf, S.; Fridell, E.; Lehtveer, M. The Potential Role of Ammonia as Marine Fuel-Based on Energy Systems Modeling and Multi-Criteria Decision Analysis. Sustain. 2020, 12, 10–14, doi:10.3390/SU12083265.
- ABS LNG as Marine Fuel; 2021;
- Boulougouris, E.K.; Chrysinas, L.E. LNG Fueled Vessels Design Training. 2015.
- Ge, J.; Wang, X. Techno-Economic Study of LNG Diesel Power (Dual Fuel) Ship. WMU J. Marit. Aff. 2017, 16, 233–245, doi:10.1007/s13437-016-0120-x.
- Ellis, J.; Tanneberger, K. Study on the Use of Ethyl and Methyl Alcohol as Alternative Fuels in Shipping; 2015;
- ABS Methanol as Marine Fuel. 2021, 1.
- MAN Diesel & Turbo Using Methanol Fuel in the MAN B&W ME-LGI Series. 2014, 1–16.
- Rachow, M.; Loest, S.; Bramastha, A.D. Analysis of the Requirement for the Ships Using Methanol as Fuel. Int. J. Mar. Eng. Innov. Res. 2018, 3, doi:10.12962/j25481479.v3i2.4054.
- Tanzer, S.E.; Posada, J.; Geraedts, S.; Ramírez, A. Lignocellulosic Marine Biofuel: Technoeconomic and Environmental Assessment for Production in Brazil and Sweden. J. Clean. Prod. 2019, 239, doi:10.1016/j.jclepro.2019.117845.
- Carvalho, F.; Portugal‐pereira, J.; Junginger, M.; Szklo, A. Biofuels for Maritime Transportation: A Spatial, Techno‐economic, and Logistic Analysis in Brazil, Europe, South Africa, and the Usa. Energies 2021, 14, doi:10.3390/en14164980.
- Brynolf, S.; Taljegard, M.; Grahn, M.; Hansson, J. Electrofuels for the Transport Sector: A Review of Production Costs. Renew. Sustain. Energy Rev. 2018, 81, 1887–1905, doi:10.1016/j.rser.2017.05.288.
- Xing, H.; Stuart, C.; Spence, S.; Chen, H. Alternative Fuel Options for Low Carbon Maritime Transportation: Pathways to 2050. J. Clean. Prod. 2021, 297, 126651, doi:10.1016/j.jclepro.2021.126651.
- Harahap, F.; Nurdiawati, A.; Conti, D.; Leduc, S.; Urban, F. Renewable Marine Fuel Production for Decarbonised Maritime Shipping: Pathways, Policy Measures and Transition Dynamics. J. Clean. Prod. 2023, 415, 137906, doi:10.1016/j.jclepro.2023.137906.
- Bengtsson, S.; Fridell, E.; Andersson, K. Environmental Assessment of Two Pathways towards the Use of Biofuels in Shipping. Energy Policy 2012, 44, 451–463, doi:10.1016/j.enpol.2012.02.030.
- Huang, J.; Fan, H.; Xu, X.; Liu, Z. Life Cycle Greenhouse Gas Emission Assessment for Using Alternative Marine Fuels: A Very Large Crude Carrier (VLCC) Case Study. J. Mar. Sci. Eng. 2022, 10, 1–17, doi:10.3390/jmse10121969.
- Bouman, E.A.; Lindstad, E.; Rialland, A.I.; Strømman, A.H. State-of-the-Art Technologies, Measures, and Potential for Reducing GHG Emissions from Shipping – A Review. Transp. Res. Part D Transp. Environ. 2017, 52, 408–421, doi:10.1016/j.trd.2017.03.022.
- Mohd Noor, C.W.; Noor, M.M.; Mamat, R. Biodiesel as Alternative Fuel for Marine Diesel Engine Applications: A Review. Renew. Sustain. Energy Rev. 2018, 94, 127–142, doi:10.1016/j.rser.2018.05.031.
- Bilgili, L. A Systematic Review on the Acceptance of Alternative Marine Fuels. Renew. Sustain. Energy Rev. 2023, 182, 113367, doi:10.1016/j.rser.2023.113367.
- Paulauskiene, T.; Bucas, M.; Laukinaite, A. Alternative Fuels for Marine Applications: Biomethanol-Biodiesel-Diesel Blends. Fuel 2019, 248, 161–167, doi:10.1016/j.fuel.2019.03.082.
- Faber, J.; Hanayama, S.; Zhang, S.; Pereda, P.; Comer, B.; Hauerhof, E.; Loeff, W.S. van der; Smith, T.; Zhang, Y.; Kosaka, H.; et al. Fourth IMO GHG Study 2020; 2020;
- Gray, N.; McDonagh, S.; O’Shea, R.; Smyth, B.; Murphy, J.D. Decarbonising Ships, Planes and Trucks: An Analysis of Suitable Low-Carbon Fuels for the Maritime, Aviation and Haulage Sectors. Adv. Appl. Energy 2021, 1, 100008, doi:10.1016/j.adapen.2021.100008.
- Xing, H.; Stuart, C.; Spence, S.; Chen, H. Fuel Cell Power Systems for Maritime Applications : Progress and Perspectives. 2021.
- Huang, Y. Conversion of a Pilot Boat to Operation on Methanol, CHALMERS UNIVERSITY OF TECHNOLOGY, 2015.
- Hansson, J.; Fridell, E.; Brynolf, S. On the Potential of Ammonia as Fuel for Shipping – A Synthesis of Knowledge. 2019.
- Kay, C.; Ng, L.; Liu, M.; Siu, J.; Lam, L.; Yang, M. Accidental Release of Ammonia during Ammonia Bunkering : Dispersion Behaviour under the Influence of Operational and Weather Conditions in Singapore. J. Hazard. Mater. 2023, 452, 131281, doi:10.1016/j.jhazmat.2023.131281.
- Fan, H.; Enshaei, H. Quantitative Risk Assessment for Ammonia Ship-to-Ship Bunkering Based on Bayesian Network. 2022, 395–410, doi:10.1002/prs.12326.
- Aneziris, O.; Koromila, I.; Nivolianitou, Z. A Systematic Literature Review on LNG Safety at Ports. Saf. Sci. 2020, 124, 104595, doi:10.1016/j.ssci.2019.104595.
- American Bureau of Shipping Propulsion and Auxiliary Systems for Gas Fuelled Ships; 2011; Vol. 2011;.
- Man Energy Solutions Engineering the Future Future in the: Two-Stroke Green-Ammonia Engine; 2019;
- Alfa Laval; Hafnia; Haldor Topsoe; Vestas; Siemens Gamesa Ammonfuel-an Industrial View of Ammonia as a Marine Fuel; 2020;
- DNV GL Part 6 Additional Class Notations Chapter 2 Propulsion, Power Generation and Auxiliary Systems. Rules Classif. Ships 2020.
- Tabibian, S.S.; Sharifzadeh, M. Statistical and Analytical Investigation of Methanol Applications , Production Technologies , Value-Chain and Economy with a Special Focus on Renewable Methanol. Renew. Sustain. Energy Rev. 2023, 179, 113281, doi:10.1016/j.rser.2023.113281.

Round 2
Reviewer 2 Report
Comments and Suggestions for Authors
All my concerns have well addressed.
Please adjust the reference format in accordance with MDPI's guidelines. Some information of the references are missing, for example, the publishers, the journal titles, etc.
Author Response
Dear Reviewer,
We sincerely thank you for your comment regarding the references. In response, we have diligently revised them in accordance with the MDPI reference style guidelines, ensuring that any missing information has been thoughtfully included. Additionally, we have identified and removed duplicate references of the manuscript.
Thank you once again for your constructive input.
